# GB83, an Agonist of PAR2 with a Unique Mechanism of Action Distinct from Trypsin and PAR2-AP

**DOI:** 10.3390/ijms231810631

**Published:** 2022-09-13

**Authors:** Yunkyung Heo, Eunhee Yang, Yechan Lee, Yohan Seo, Kunhi Ryu, Hyejin Jeon, Wan Namkung

**Affiliations:** 1College of Pharmacy, Yonsei Institute of Pharmaceutical Sciences, Yonsei University, Incheon 21983, Korea; 2New Drug Development Center, Daegu Gyeongbuk Medical Innovation Foundation, Daegu 41061, Korea

**Keywords:** PAR2, GB83, agonist, desensitization

## Abstract

Protease-activated receptor 2 (PAR2) is a G-protein-coupled receptor (GPCR) activated by proteolytic cleavage of its N-terminal domain. Once activated, PAR2 is rapidly desensitized and internalized by phosphorylation and β-arrestin recruitment. Due to its irreversible activation mechanism, some agonists that rapidly desensitized PAR2 have been misconceived as antagonists, and this has impeded a better understanding of the pathophysiological role of PAR2. In the present study, we found that GB83, initially identified as a PAR2 antagonist, is a bona fide agonist of PAR2 that induces unique cellular signaling, distinct from trypsin and PAR2-activating peptide (AP). Activation of PAR2 by GB83 markedly elicited an increase in intracellular calcium levels and phosphorylation of MAPKs, but in a delayed and sustained manner compared to the rapid and transient signals induced by trypsin and PAR2-AP. Interestingly, unlike PAR2-AP, GB83 and trypsin induced sustained receptor endocytosis and PAR2 colocalization with β-arrestin. Moreover, the recovery of the localization and function of PAR2 was significantly delayed after stimulation by GB83, which may be the reason why GB83 is recognized as an antagonist of PAR2. Our results revealed that GB83 is a bona fide agonist of PAR2 that uniquely modulates PAR2-mediated cellular signaling and is a useful pharmacological tool for studying the pathophysiological role of PAR2.

## 1. Introduction

Protease-activated receptor 2 (PAR2), also known as thrombin receptor-like 1 (F2RL1), is a G-protein-coupled receptor (GPCR) expressed in many cell types, such as immune cells and epithelial cells of various tissues, including the gastrointestinal tract, lungs, and skin [1,2,3]. PAR2 is activated by proteolytic cleavage of the N-terminus by extracellular serine proteases such as trypsin, tryptase, and neutrophil elastase, unmasking self-activating endogenous tether ligands [4]. Activation of PAR2 induces multiple G-protein-mediated signaling pathways, which are involved in a wide range of physiological responses such as inflammation, cell proliferation, and angiogenesis [4,5,6].

Activation of PAR2 triggers intracellular calcium increase through phospholipase C/Ca2+/protein kinase C signaling, along with the activation of NF-κB signaling pathways [7]. Downstream of these signaling pathways leading to the production of inflammatory cytokines initiates the PAR2-mediated inflammatory axis [8,9]. Research over the past two decades has revealed the critical roles of PAR2 in various inflammatory diseases such as arthritis, atopic dermatitis, asthma, colitis, and inflammatory bowel disease [10,11,12,13,14]. Moreover, a recent protein–protein interaction study identified the direct interaction between PAR2 and SARS-CoV-2 viral protein, ORF9c [15], which suggests that PAR2 is one of the druggable potential targets to treat severe COVID-19-associated hyperinflammation.

In response to activation of PAR2 by ligands, PAR2 undergoes rapid desensitization, which requires receptor phosphorylation by G-protein-coupled regulatory kinases (GRKs) that provides binding sites to β-arrestins [16]. It is well established that recruited β-arrestins regulate uncoupling from G proteins and endocytosis of activated PAR2 [17]. Proteolytic activation of PAR2 by trypsin induces rapid translocation of β-arrestins from the cytosol to the plasma membrane forming PAR2/β-arrestin complexes, followed by association with clathrin-coated vesicles for endocytosis [18]. Internalized PAR2 remains stably associated with β-arrestins in endosomes, consequently degraded in lysosomes, and slowly recovers its distribution at the plasma membrane. Because resensitization of irreversibly activated PAR2 requires a long process, several compounds that actually induce activation of PAR2, along with receptor desensitization have been misrecognized as PAR2 antagonists, such as K-14585, C391, and GB88 [19,20,21]. In this study, surprisingly, we found that GB83, developed as a PAR2 selective antagonist [22], is a bona fide agonist of PAR2. Interestingly, although the signaling mechanisms induced by GB83 have not been clearly elucidated, several previous studies have shown that GB83 attenuates a wide range of PAR2-mediated inflammatory responses [23,24,25,26,27]. Moreover, our previous study showed that GB83 acts as a positive allosteric modulator of PAR1 [28].

The lack of a clear understanding of the pharmacological properties and mechanism of action of GB83 may impede a better understanding of the pathophysiological role of PAR2. Therefore, in this study, we investigated the PAR2 signaling pathway triggered by GB83 in comparison to that induced by trypsin and PAR2-AP.

## 2. Results

### 2.1. GB83 Is a Bona Fide PAR2 Agonist and Induces Elevation of Intracellular Calcium through Activation of PAR2

GB83 was developed as a non-peptide antagonist that reversibly inhibits PAR2 at low micromolar concentrations [22]. Consistent with the previous report, pretreatment of GB83 potently inhibited PAR2-AP-induced intracellular calcium increase in human colorectal adenocarcinoma HT-29 cells with IC_50_ of 2.1 ± 0.45 µM (Figure 1A,B). However, unexpectedly, when we treated cells with GB83 and observed the changes in intracellular calcium concentrations, we found that GB83 is not an antagonist of PAR2 but an agonist of PAR2. To investigate the characteristics of GB83 as a PAR2 agonist, GB83 was compared with two major PAR2 agonists, trypsin, the predominant endogenous protease that activates PAR2, and PAR2-AP, a synthetic peptide that selectively activates PAR2. As shown in Figure 1C–E, trypsin, GB83, and PAR2-AP induced intracellular calcium increases in HT-29 cells in a dose-dependent manner. In contrast to a rapid and transient calcium spike triggered by both trypsin and PAR2-AP, GB83 weakly and slowly induced calcium increase. In addition, trypsin and PAR2-AP reached a calcium peak within 20 s after application, whereas GB83 reached a calcium peak over 60 s after application. It is noteworthy that trypsin-, GB83- and PAR2-AP-induced intracellular calcium increases were almost completely blocked by pretreatment of AZ3451, a potent and selective antagonist of PAR2 with IC_50_ of 23 nM [29]. These results suggest that GB83 induces elevation of intracellular calcium through activation of PAR2.

To clearly elucidate PAR2 activation by GB83, additional experiments were performed using human melanoma A2058 cells deficient in endogenous expression of PAR2. As shown in Figure 2A, application of PAR2-AP and GB83 did not change the intracellular calcium concentration in A2058 cells, but PAR1-AP treatment resulted in a rapid and robust increase in intracellular calcium concentration. To investigate the effect of GB83 on human PAR2, A2058 cells were stably transfected with human PAR2. As shown in Figure 2B, PAR2-AP strongly increased intracellular calcium levels in a dose-dependent manner in A2058-PAR2 cells. Notably, GB83 also markedly elicited intracellular calcium increase in A2058-PAR2 cells in a dose-dependent manner (EC_50_ = 2.1 ± 0.17 µM). Both PAR2-AP and GB83-induced increases in intracellular calcium levels were completely blocked by 1 μM of AZ3451 (Figure 2B,C). These results clearly show that GB83 is a bona fide agonist of PAR2.

### 2.2. GB83 Induces PAR2-Mediated Phosphorylation of ERK1/2 and p38

Activation of MAPKs including p42/44 (ERK1/2) and p38 is a typical signaling pathway triggered by PAR2 activation. It is well known that PAR2 promotes cell proliferation and migration via ERK1/2 and p38 MAPK signaling pathways [30,31]. To investigate the effect of GB83 on PAR2-mediated MAPK signaling compared to trypsin and PAR2-AP, HT-29 cells were treated with trypsin, GB83, and PAR2-AP. Trypsin, GB83, and PAR2-AP all significantly increased phosphorylation of ERK1/2 (Figure 3A,B). To determine whether phosphorylation of ERK1/2 by GB83 is mediated by a PAR2-mediated signaling pathway, HT-29 cells were pretreated with AZ3451. As shown in Figure 3C,D, the phosphorylation of ERK1/2 by GB83 was completely inhibited by AZ3451. In the case of p38 MAPK, GB83, trypsin, and PAR2-AP significantly stimulated phosphorylation of p38, but GB83 showed longer activation of p38 compared to trypsin and PAR2-AP (Figure 4A,B). GB83-induced phosphorylation of p38 MAPK was significantly blocked by AZ3451 (Figure 4C,D). These findings indicate that GB83 triggers phosphorylation of ERK1/2 and p38 MAPK via PAR2 activation.

### 2.3. GB83 Induces Sustained Endocytosis of PAR2

It is well established that agonist-induced activation of PARs leads to receptor internalization. To investigate the effect of GB83 on internalization of PAR2, HT-29 cells were stably transfected with EGFP-tagged PAR2 or PAR4, and agonist-induced receptor internalization was examined in live cells using an automated fluorescence microscopy system. Trypsin, GB83, and PAR2-AP significantly induced internalization of PAR2, but PAR4-AP did not induce PAR2 internalization (Figure 5A). The extent of receptor internalization induced by these agonists was quantitatively compared by measuring the number of GFP puncta. Notably, PAR2 internalization induced by trypsin and GB83 was maintained for a longer period of time, whereas that induced by PAR2-AP was rapidly decreased (Figure 5B). In the case of PAR4 internalization, GB83 and PAR2-AP did not affect internalization of PAR4 (Figure 5C). However, trypsin induced internalization of PAR4 like PAR4-AP, but this result is not surprising, as previous studies have shown that thrombin and trypsin can induce activation of PAR4 [32]. The extent of PAR2 and PAR4 internalization induced by agonists was further analyzed through quantitative measurement of fold change in the number of GFP puncta (Figure 5D). These results indicated that GB83 and PAR2-AP selectively induce internalization of PAR2.

### 2.4. GB83 Induces Sustained Colocalization of PAR2 and β-Arrestins

It is well known that β-arrestins bind to activated GPCRs and regulate their desensitization, signal attenuation, and receptor trafficking [17]. We examined the effect of trypsin, GB83, and PAR2-AP on translocation of EGFP-tagged PAR2 and its colocalization with RFP-tagged β-arrestin1/2. To examine the colocalization of PAR2 and β-arrestin1/2, HT-29 cells were co-transfected with EGFP-tagged PAR2 and RFP-tagged β-arrestin1 or β-arrestin2. Interestingly, trypsin- and GB83-induced internalized PAR2 demonstrated colocalization with β-arrestin1, whereas PAR2-AP-induced internalized PAR2 showed less overlap with β-arrestin1 (Figure 6). The extent of colocalization between PAR2 and β-arrestin1 was precisely quantified as Pearson’s correlation coefficient (PCC). As a result, a high degree of colocalization was induced by trypsin and GB83, whereas a low level of colocalization between PAR2 and β-arrestin1 was induced by PAR2-AP (Figure 6D). The colocalization of PAR2 and β-arrestin2 also showed similar results to β-arrestin1 (Figure 7). These results suggest that GB83 and trypsin share the capacity of inducing sustained colocalization of internalized PAR2 and β-arrestins.

### 2.5. GB83 Induces Prolonged PAR2 Desensitization and Slow Recovery of Cell Surface PAR2

Since several studies have provided evidence for the inhibitory effect of GB83 on PAR2 [23,24,25,26,27], we attempted to investigate the possibility that GB83 interferes with recovery of cell surface PAR2. First, we observed the recovery of PAR2 on the plasma membrane after initial stimulation by trypsin, GB83, and PAR2-AP in HT-29 cells stably expressing EGFP-tagged PAR2. Interestingly, GB83 showed a more delayed recovery of PAR2 on the plasma membrane compared to trypsin and PAR2-AP. After 12 h of stimulation, trypsin and PAR2-AP treated cells restored PAR2 to pre-stimulation levels, but internalized PAR2 was still largely observed in GB83 treated cells (Figure 8A). Next, we investigated the functional recovery of PAR2 in HT-29 cells after initial stimulation with trypsin, GB83, and PAR2-AP. As shown in Figure 8B–E, non-enzymatic activation of PAR2 by PAR2-AP showed a significantly faster recovery of cell surface PAR2 compared to trypsin and GB83 3 h after stimulation. Notably, GB83 showed significantly slower recovery of cell surface PAR2 compared to trypsin and PAR2-AP 6 h after stimulation. These results showed that GB83 induces delayed functional recovery of PAR2, and this effect of GB83 on PAR2 suggests that long-term use of GB83 may exert an inhibitory effect on PAR2.

## 3. Discussion

PAR2 is activated by several serine proteases, including trypsin and tryptase that induce N-terminal proteolytic cleavage, and synthetic agonist peptides, and the activated PAR2 couples to G proteins to trigger a variety of cellular responses. Once activated, PAR2 is phosphorylated by G-protein-coupled receptor kinases (GRKs), and β-arrestin binds to phosphorylated PAR2, triggering desensitization and internalization of PAR2 [17]. Since internalized PAR2 is degraded in lysosomes, resensitization occurs through recovery of PAR2 at the plasma membrane by intracellular storage mobilization and de novo synthesis of PAR2 [33]. The unique regulatory mechanisms of PAR2 have hindered the development of PAR2 modulators. For example, PAR2 modulators such as K-14585, C391, and GB88 were initially identified as PAR2 antagonists but were found to exhibit partial agonist activity [19,20,21]. The lack of a clear distinction between PAR2 agonists and antagonists has hampered a better understanding of the pathophysiological role of PAR2. In this study, we evaluated the properties of GB83, initially identified and widely used as a potent PAR2 antagonist, and found that GB83 is a bona fide agonist of PAR2 that induces a PAR2-mediated cellular response distinct from trypsin and PAR2-AP.

GB83 was identified as the first non-peptidyl antagonist of PAR2 that reversibly inhibits PAR2 at low micromolar concentrations (IC_50_ = 2 µM) [22]. In HT-29 cells, intracellular calcium increase by PAR2-AP was potently inhibited by pretreatment of GB83 with an IC_50_ of 2.1 μM, which is consistent with the results of Barry et al. (Figure 1A,B). Because there are cases in which PAR2 modulators acting as PAR2 agonists have been identified among PAR2 antagonists [19,20,21], we observed whether GB83 activates PAR2. Surprisingly, GB83 showed a significant increase in calcium concentration, although lower than the increase in intracellular calcium levels by trypsin and PAR2-AP, and the GB83-induced intracellular calcium increase was completely blocked by AZ3451, a selective PAR2 antagonist (Figure 1C–E). In addition, GB83 did not induce intracellular calcium increase in A2058 cells not expressing functional PAR2 but strongly increased intracellular calcium levels in A2058 cells overexpressing human PAR2 (Figure 2). These results clearly revealed that GB83 induces intracellular calcium increase through PAR2 activation.

Trypsin and PAR2-AP rapidly induced strong and transient calcium signals, whereas GB83 induced a slow and prolonged calcium signal (Figure 1C–E and Appendix A). As shown in Appendix AB,C, when intracellular calcium signaling via PAR2 activation by PAR2-AP and GB83 was observed at the single-cell level, the increase in intracellular calcium levels induced by GB83 was similar to that of PAR2-AP. However, PAR2-AP induced PAR2 activation simultaneously in most cells, whereas GB83 induced asynchronous PAR2 activation. In this study, we did not elucidate the underlying mechanism of GB83-induced asynchronous PAR2 activation, but this phenomenon may be attributed to the pharmacological properties of GB83, such as membrane permeability and solubility, or the binding mechanism between GB83 and PAR2.

Interestingly, GB83 has been used as a PAR2 antagonist in several in vivo studies and exhibited PAR2 inhibitory effects similar to those seen in PAR2 KO mice. For example, GB83 significantly reduced the increase in vascular perfusion and rolling leukocytes in neutrophil elastase-induced joint inflammation, similar to PAR2 KO [24]. The reduction of oxidative stress and inflammation for UVB-induced skin photoaging in PAR2 KO mice was also shown in normal mice treated with GB83 [24]. In inflammation in a high-fat diet environment, proinflammatory cytokine levels were significantly increased in PAR2 KO mice and GB83-treated normal mice compared to wild-type mice and vehicle-treated normal mice, respectively [34]. In the present study, we showed that GB83 induces sustained endocytosis of PAR2 to similar levels elicited by trypsin (Figure 5), but both locational and functional recovery of PAR2 was markedly delayed after stimulation by GB83 (Figure 8). In addition, brefeldin A treatment strongly blocked the restoration of cell membrane expression of PAR2 from GB83, PAR2-AP, and trypsin-induced endocytosis of PAR2 (Appendix A). These results suggest that the recovery of PAR2 from GB83-induced endocytosis is mainly due to the neo-synthesis of PAR2. Although the underlying mechanism for the delayed recovery has not been fully understood, these results suggest that long-term treatment of GB83 may exert PAR2 inhibitory effects by inducing receptor desensitization and endocytosis and interfering with resensitization, providing a rationale for why GB83 exhibits PAR2 inhibitory effects in vivo.

β-Arrestin is a key component linking GPCRs to the endocytic machinery, and the stability of the GPCR/β-arrestin complex determines the trafficking of receptors that are rapidly redistributed to the plasma membrane or degraded and slowly recovered [35,36]. We visualized the interaction of PAR2 with β-arrestins and found important differences between the two synthetic PAR2 agonists, GB83 and PAR2-AP. While GB83 markedly induced colocalization of PAR2 and β-arrestin, PAR2-AP elicited a low level of colocalization of the two proteins (Figure 6 and Figure 7). These results suggest that GB83 is a PAR2 agonist with different properties from PAR2-AP and successfully induces both G-protein-mediated signaling and β-arrestin recruitment. Notably, β-arrestins have recently been recognized as an activator of G-protein-independent signaling involved in various physiological processes [37]. For example, β-arrestin2 acts as an important regulator of the PAR2-induced inflammatory response in asthma [38]. Given the role of β-arrestin in the PAR2-mediated inflammatory responses, GB83 could be a useful pharmacological tool to investigate the role of β-arrestin signaling by PAR2 in several inflammatory diseases.

In summary, we found that GB83 is an agonist of PAR2. GB83 markedly elicited sustained intracellular calcium increase, MAPK phosphorylation, β-arrestin recruitment, and receptor endocytosis. Moreover, compared with PAR2-AP and trypsin, GB83 particularly induced a delay in cell surface recovery of PAR2, which provides a rationale for why GB83 has exhibited PAR2 inhibitory effects in previous in vitro and in vivo studies. Taken together, GB83 has a different mode of action compared to trypsin and PAR2-AP, so it can be usefully used to investigate the pathophysiological roles of PAR2 in various diseases.

## 4. Materials and Methods

### 4.1. Cell Culture and Cell Lines

Human colorectal adenocarcinoma (HT-29) and human metastatic melanoma (*A2058*) cells were cultured at 37 °C and 5% CO2. *A2058* cells were grown in DMEM high-glucose medium (Welgene Inc., Gyeongsan, Korea), and HT-29 cells were grown in RPMI1640 medium (Welgene Inc., Gyeongsan, Korea). All media were supplemented with 10% FBS, 100 units/mL penicillin, and 100 µg/mL streptomycin, and all cells were purchased from the Korean Cell line Bank (Seoul, Korea).

### 4.2. Materials and Reagents

Trypsin was purchased from Sigma-Aldrich (St. Louis, MO, USA), AZ3451 from Tocris Bioscience (Atlantic Road, Bristol, UK), and GB83 from Axon Medchem (Groningen, The Netherlands). PAR2-AP (SLIGRL-NH2) was synthesized from Cosmogenetech Co., Ltd. (Seoul, Korea).

### 4.3. Molecular Cloning of Plasmid Constructs

The PAR2 coding sequence was amplified by polymerase chain reaction (PCR) using PAR2 plasmid (Genbank Accession No. NM_005252.5) purchased from OriGene Technologies (Rockville, MD, USA) as the template (forward primer: TTTTT GAATTC CACC ATG CGG AGC CCC AGC, reverse primer: TTTTT TCTAGA TTA ATA GGA GGT CTT AAC AGT GG). The PCR products were digested using EcoR1 and Xba1 restriction enzymes (Enzynomics, Daejeon, Korea) and inserted into pLVX-EF1α-IRES-puro (pLVX-EIP) vector (a kind gift from Professor Jinu Lee).

The PAR2-EGFP construct was generated by inserting the EGFP coding sequence (forward primer: TTTTA TCTAGA GGA GGA AGC AAG GGC GAG GAG, reverse primer: TTTTT GGATCC TTA CTT GTA CAG CTC GTCC) to the pLVX-EIP vector using Xba1 and BamH1 as restriction enzymes and, subsequently, the PAR2 coding sequence with a stop codon substituted with a codon encoding glycine (forward primer: TTTTT GAATTC CACC ATG CGG AGC CCC AGC, reverse primer: TTTTT TCTAGA GCC ATA GGA GGT CTT AAC AGT GG) digested with EcoR1 and Xba1. The PAR4-EGFP construct was generated likewise using PAR4 plasmid (Genbank Accession No. NM_003950.4) purchased from Sino Biological (Beijing, China) as the template (forward primer: TTTTT G AATTC ACC ATG TGG GGG CGA CTGC, reverse primer: TTTTT TCTAGA ACC CTG GAG CAA AGA GGA GTG).

The BARR1-mCherry and BARR2-mCherry constructs were generated by the inserting mCherry coding sequence into the pLVX-EF1α-IRES-Bla (pLVX-EIBla) vector, a kind gift from Professor Jinu Lee (forward primer: TTTTT ACTAGT GGA GGA AGC AAG GGC GAA GAGG, reverse primer: TTTTT GAATTC TTA CTT GTA CAG CTC GTCC) digested with Spe1 and BamH1, and subsequently β-arrestin1 and β-arrestin2 coding sequences, respectively. Templates for β-arrestin1/2 were purchased from the Korea Human Gene Bank (Daejeon, Korea). All inserted regions were sequenced by Bionics Co., Ltd. (Seoul, Korea).

### 4.4. Generation of Stable Cell Line

All stable cell lines were generated through lentiviral transduction. Lentivirus was produced by transfecting generated transfer plasmids to HEK293T cells with packaging and envelope plasmids, psPAX2 (Addgene Plasmid No. 12260) and pMD2.G (Addgene Plasmid No. 12259). Transfection was performed with Lipofectamine 3000 transfection reagent (Thermo Scientific, Waltham, MA, USA), following the manufacturer’s instruction. Transfected cells were incubated at 37 °C and viral supernatants were harvested at 48 h and 72 h post transfection and filtered with a 0.45 μm PES filter. A mixture of viral supernatant and culture media with a ratio of 1:1 was applied to HT-29 cells, and successfully transduced cells were selected by applying puromycin (2 μg/mL) or blasticidin (20 μg/mL) 72 h post transduction.

### 4.5. Intracellular Calcium Measurement

HT-29 and A-2058 cells were grown on 96-well clear-bottom black wall plates (Corning Inc., Corning, NY, USA). The intracellular calcium was measured using the Fluo-4 NW kit (Invitrogen, Carlsbad, CA, USA) following the manufacturer’s instructions. Briefly, cells were incubated with 100 µL assay buffer with Fluo-4 dye for 1 h. The fluorescence was measured using the FLUOstar Omega microplate reader (BMG LABTECH, Offenburg, Germany).

### 4.6. Immunoblotting

For Western blot analysis, HT-29 cells were plated on 6-well-plates and incubated overnight. Cells were treated with compounds accordingly, washed twice with ice-cold PBS, and lysed for 15 min in RIPA buffer supplemented with a protease inhibitor cocktail. Lysed samples were centrifuged at 13,000 rpm for 20 min at 4 °C. Extracted proteins were quantified using the Bradford protein assay kit (Thermo Scientific, Waltham, MA, USA), and 30 µg of total proteins was loaded to each well and separated by 4–12% Tris-glycine precast gel (Koma Biotech, Seoul, Korea). Proteins were transferred to PVDF membranes (Millipore, Billerica, MA, USA), followed by blocking for 1 h with 5% BSA in Tris-buffered saline with 0.1% Tween-20 (TBST). The membranes were incubated with primary antibodies overnight at 4 °C with the indicated primary antibodies; anti-p38 (Santa Cruz Biotechnologies, Santa Cruz, CA, USA, Cat. No. sc-7972, RRID: AB_628079), anti-phospho-p38 (Santa Cruz Biotechnologies, Santa Cruz, CA, USA, Cat. No. sc-166182, RRID: AB_2141746), anti-p42/44 (Cell Signaling, Danvers, MA, USA, Cat. No. 9102, RRID: AB_330744), anti-phospho-p42/44 (Cell Signaling, Danvers, MA, USA, Cat. No. 9101, RRID: AB_331646), and anti-β-actin (Santa Cruz Biotechnologies, Santa Cruz, CA, USA, Cat. No. sc-47778, RRID: AB_626632). Then, the membranes were washed three times in TBST and incubated with horseradish peroxidase-conjugated secondary antibodies for 1 h. After being washed three times, membranes were detected using the ECL Plus immunoblotting detection system (GE Healthcare, Piscataway, NJ, USA). All experiments were repeated 5 times independently, and ImageJ software (NIH, Bethesda, MD, USA) was used for result analysis.

### 4.7. Live Cell Imaging

Cells were grown overnight on 96-well clear-bottom black wall plates (Corning Inc., Corning, NY, USA). Culture media were aspirated and exchanged with HEPES-buffered solution. Live cell images were taken using the BioTek Lionheart FX automated microscope (Agilent, Santa Clara, CA, USA). Beacons were used to define specific x/y offsets for imaging, and the specific regions were observed before and after stimulation. Number of puncta was automatically measured using Lionheart analysis software.

### 4.8. Quantification of Colocalization

Colocalization analysis of PAR2-EGFP and β-arrestin1/2-mCherry was performed using ImageJ. Colocalization values were calculated in Pearson’s correlation coefficient from 5 representative images using the colocalization plugin Just Another Colocalization Plugin (JACoP).

### 4.9. Data and Statistical Analysis

For all statistically analyzed studies, experiments were performed at least five times independently. The experiments were carried out in a randomized manner. The results are presented as the mean ± SEM. Statistical analysis was performed with one-way analysis of variance (ANOVA), followed by Tukey’s multiple comparison test for post-hoc analysis. A value of *p* < 0.05 was considered statistically significant. Concentrations of response curves were fitted in GraphPad Prism 5.0 (GraphPad software, San Diego, CA, USA).

## Figures and Tables

**Figure 1 ijms-23-10631-f001:**
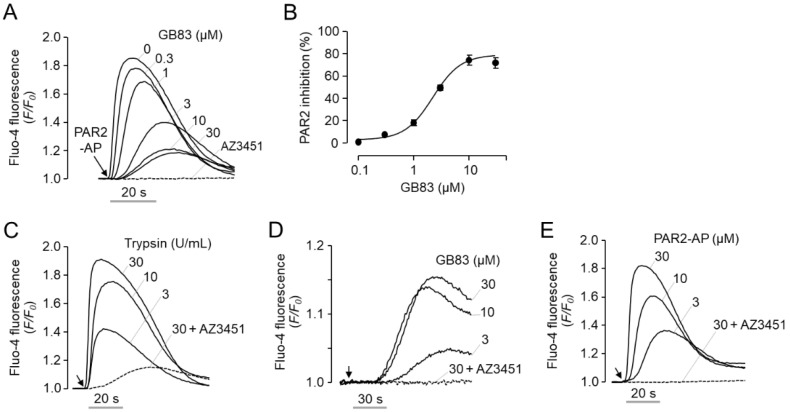
Effect of GB83 on PAR-2-mediated increase in intracellular calcium levels in HT-29 cells. (**A**) Inhibitory effect of GB83 on PAR2-mediated intracellular calcium increase. HT-29 cells were pretreated with GB83 at indicated concentrations and AZ3451 (1 µM) for 15 min before PAR2 stimulation by PAR2-AP (30 µM). (**B**) Summary of dose–responses (mean ± S.E., *n* = 6). (**C**–**E**) Intracellular calcium levels were increased by the indicated concentrations of trypsin, GB83, and PAR2-AP in HT-29 cells. Intracellular calcium increase induced by trypsin (30 U/mL), GB83 (30 µM), and PAR2-AP (30 µM) were inhibited by 1 µM of AZ3451, a potent and selective PAR2 antagonist.

**Figure 2 ijms-23-10631-f002:**
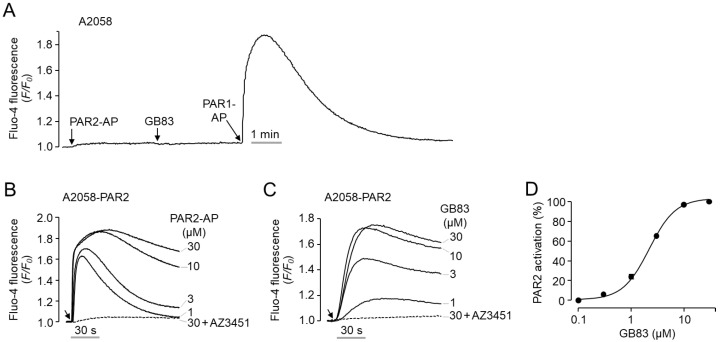
GB83 potently induced a PAR2-mediated increase in intracellular calcium concentration. (**A**) Effect of PAR2-AP (30 µM), GB83 (30 µM), and PAR1-AP (30 µM) on intracellular calcium levels in A2058 cells not expressing PAR2. (**B**,**C**) Effect of PAR2-AP and GB83 on PAR2-mediated intracellular calcium increase in A2058 cells stably transfected with human PAR2. Arrowheads indicate when PAR2-AP and GB83 were applied. AZ3451 (1 µM) was pretreated 15 min before application of PAR2-AP and GB83. (**D**) Dose–response curve for GB83-induced PAR2 activation (mean ± S.E., *n* = 5).

**Figure 3 ijms-23-10631-f003:**
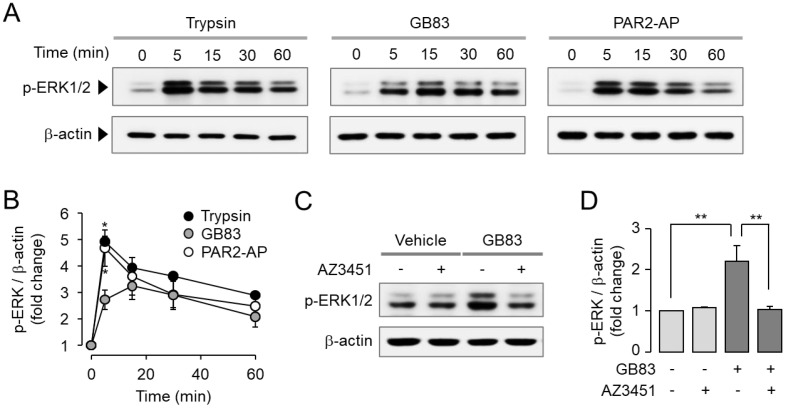
GB83 induced phosphorylation of ERK1/2 through PAR2 activation in HT-29 cells. (**A**) Representative immunoblots of ERK1/2 phosphorylation. PAR2 stimulated by trypsin (30 U/mL), GB83 (30 µM), and PAR2-AP (30 µM). (**B**) p-ERK1/2 band intensity was normalized to β-actin (mean ± S.E., *n* = 5). Statistical significance of differences was assessed by one-way ANOVA with Tukey’s post-hoc test. * *p* < 0.05, Trypsin and PAR2-AP versus GB83. (**C**) PAR2-mediated phosphorylation of ERK1/2 by GB83. AZ3451 (1 µM) was pretreated 15 min prior to stimulation with GB83 (30 µM). (**D**) p-ERK1/2 band intensity was normalized to β-actin (mean ± S.E., *n* = 5). Statistical significance of differences between indicated groups was assessed by one-way ANOVA with Tukey’s post-hoc test. ** *p* < 0.01.

**Figure 4 ijms-23-10631-f004:**
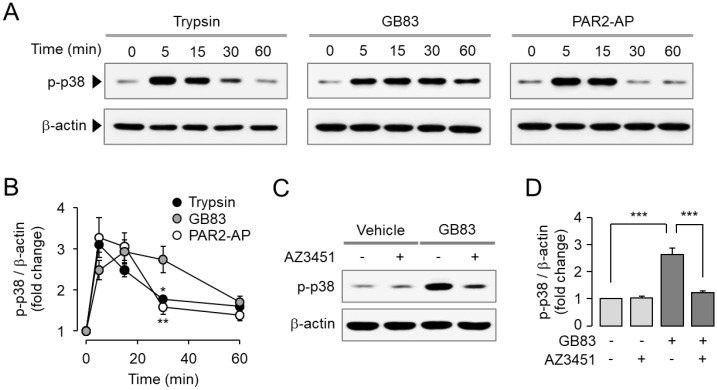
GB83 induced phosphorylation of p38 MAPK through PAR2 activation in HT-29 cells. (**A**) Representative immunoblots of p38 phosphorylation. PAR2 stimulated by trypsin (30 U/mL), GB83 (30 µM), and PAR2-AP (30 µM). (**B**) p-p38 band intensity was normalized to β-actin (mean ± S.E., *n* = 5). Statistical significance of differences was assessed by one-way ANOVA with Tukey’s post-hoc test. * *p* < 0.05 and ** *p* < 0.01 compared to GB83. (**C**) PAR2-mediated phosphorylation of p38 by GB83. AZ3451 (1 µM) was pretreated 15 min prior to stimulation with GB83 (30 µM). (**D**) p-p38 band intensity was normalized to β-actin (mean ± S.E., *n* = 5). Statistical significance of differences between indicated groups was assessed by one-way ANOVA with Tukey’s post-hoc test. *** *p* < 0.001.

**Figure 5 ijms-23-10631-f005:**
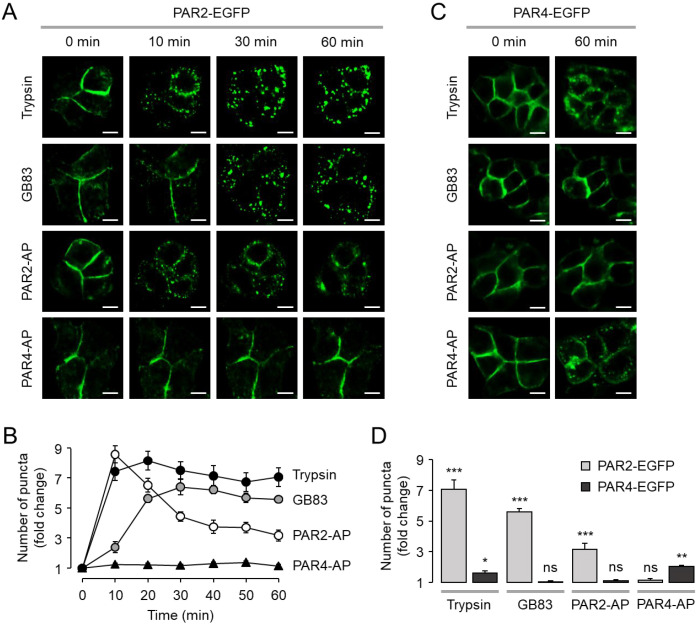
GB83 selectively and strongly internalized PAR2 in HT-29 cells. (**A**) EGFP-tagged PAR2 expressing HT-29 cells were treated with trypsin (30 U/mL), GB83 (30 µM), PAR2-AP (30 µM), and PAR4-AP (100 µM). Changes in cellular localization of EGFP-tagged PAR2 were observed at the indicated time points. (**B**) Quantitative analysis of the effect of trypsin, GB83, PAR2-AP, and PAR4-AP on PAR2 internalization. Summary of fold change in the number of puncta (mean ± S.E., *n* = 5). (**C**) EGFP-tagged PAR4 expressing HT-29 cells were treated with trypsin (30 U/mL), GB83 (30 µM), PAR2-AP (30 µM), and PAR4-AP (100 µM). Changes in cellular localization of EGFP-tagged PAR4 were observed at the indicated time points. (**D**) Quantitative analysis of the effect of trypsin, GB83, PAR2-AP, and PAR4-AP on PAR2 and PAR4 internalization. Summary of fold change in the number of puncta at 60 min (mean ± S.E., *n* = 5). Statistical significance of differences between indicated group with that before treatment was assessed by one-way ANOVA with Tukey’s post-hoc test. * *p* < 0.05, ** *p* < 0.01, *** *p* < 0.001, ns, not significant.

**Figure 6 ijms-23-10631-f006:**
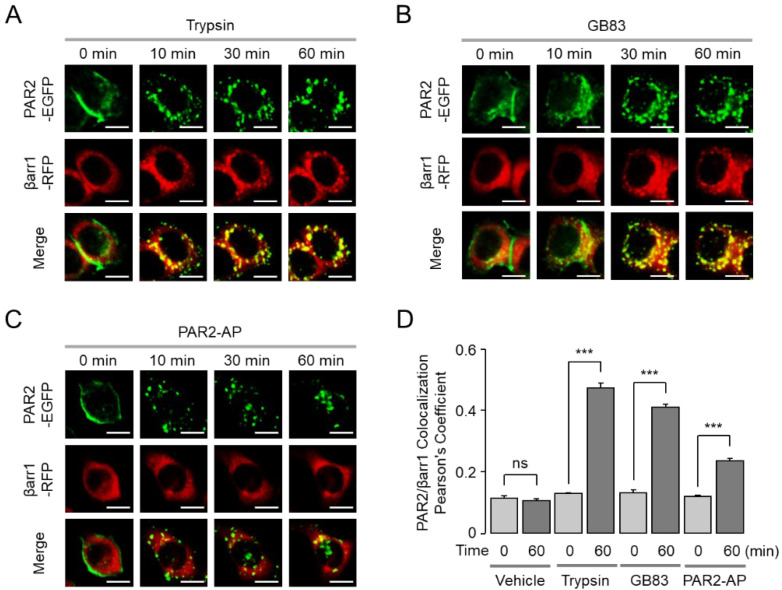
GB83 induces sustained colocalization of PAR2 and β-arrestin1. (**A**–**C**) Effect of trypsin, GB83, and PAR2-AP on the localization of PAR2 and β-arrestin1 (βarr1) in HT-29 cells expressing EGFP-tagged PAR2 and RFP-tagged β-arrestin1. Cells were treated with trypsin (30 U/mL), GB83 (30 µM), PAR2-AP (30 µM), and the cellular localization of PAR2 (green) and β-arrestin1 (red) were observed at 0, 10, 30, and 60 min after treatment of each agonist. (**D**) Quantitative analysis of colocalization of PAR2 and β-arrestin1 (mean ± S.E., *n* = 5). Colocalization values were calculated in Pearson’s correlation coefficient using the colocalization plugin JACoP. Statistical significance of differences between indicated groups was assessed by one-way ANOVA with Tukey’s post-hoc test. *** *p* < 0.001, ns, not significant.

**Figure 7 ijms-23-10631-f007:**
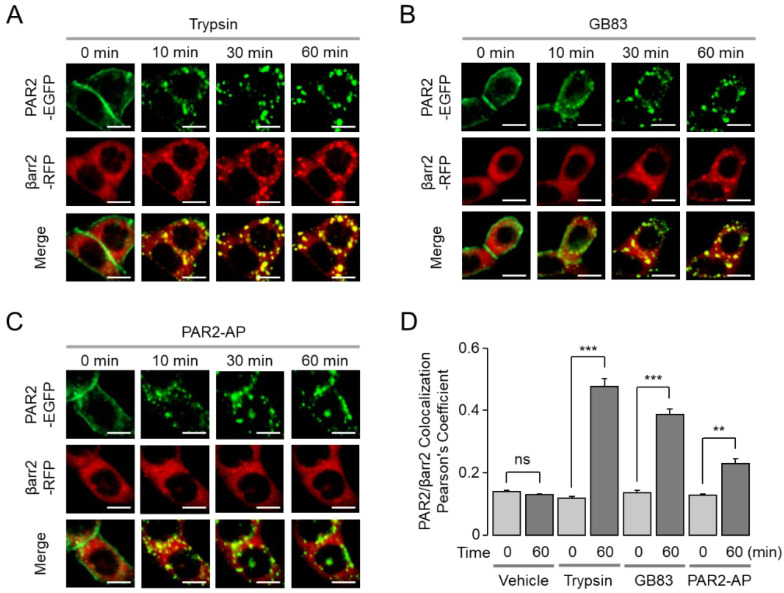
GB83 induces sustained colocalization of PAR2 and β-arrestin2. (**A**–**C**) Effect of trypsin, GB83, and PAR2-AP on the localization of PAR2 and β-arrestin2 (βarr2) in HT-29 cells expressing EGFP-tagged PAR2 and RFP-tagged β-arrestin2. Cells were treated with trypsin (30 U/mL), GB83 (30 µM), PAR2-AP (30 µM), and the cellular localization of PAR2 (green) and β-arrestin2 (red) were observed at 0, 10, 30, and 60 min after treatment of each agonist. (**D**) Quantitative analysis of colocalization of PAR2 and β-arrestin2 (mean ± S.E., *n* = 5). Colocalization values were calculated in Pearson’s correlation coefficient using the colocalization plugin JACoP. Statistical significance of differences between indicated groups was assessed by one-way ANOVA with Tukey’s post-hoc test. ** *p* < 0.01, *** *p* < 0.001, ns, not significant.

**Figure 8 ijms-23-10631-f008:**
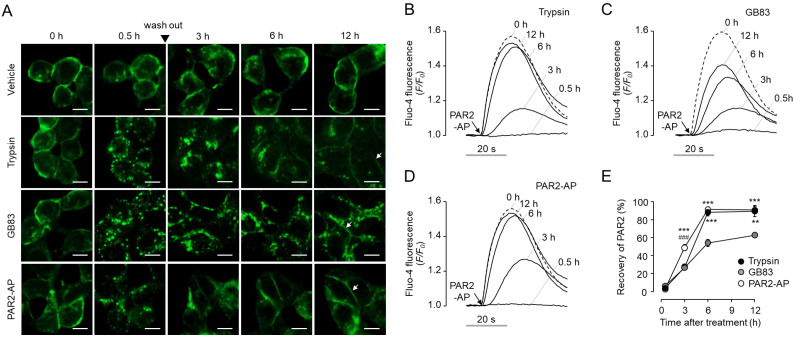
GB83 induced prolonged PAR2 desensitization and slow recovery of PAR2 in HT-29 cells. (**A**) Effect of trypsin, GB83, and PAR2-AP on cellular localization of EGFP-tagged PAR2 were observed at the indicated time points. Cells were treated with trypsin (30 U/mL), GB83 (30 μM) or PAR2-AP (30 μM) for 30 min, washed 3 times with PBS, and then replaced with culture medium. (**B**–**D**) Effect of trypsin, GB83, and PAR2-AP on functional recovery of PAR2. HT-29 cells were treated with trypsin (30 U/mL), GB83 (30 µM), or PAR2-AP (30 µM) for 30 min, washed 3 times with PBS, and then replaced with culture medium. PAR2-AP-mediated increases in intracellular calcium levels were measured at the indicated time points after wash out. (**E**) Summary of functional recovery of PAR2 (mean ± S.E., *n* = 5). Statistical significance of differences was assessed by one-way ANOVA with Tukey’s post-hoc test. ** *p* < 0.01 and *** *p* < 0.001 compared to GB83; ### *p* < 0.001 compared to trypsin.

## Data Availability

Not applicable.

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
