# Peer review of "GB83, an Agonist of PAR2 with a Unique Mechanism of Action Distinct from Trypsin and PAR2-AP"

_ijms, 2022, doi:10.3390/ijms231810631_

Round 1
Reviewer 1 Report
This paper shows, that contrary to previous reports in the literature, GB83 is a rapidly inactivating agonist at the PAR2. The data is novel and convincing, although the statistical analysis may need attention. I feel the paper will be of interest, given that the compound is routinely described as an antagonist in the literature.
1) The authors state that they use a paired t-test. This is appropriate for most of their experiments, where they are simply comparing two variables; the t-test cannot be used to compare three or more values. However, I am puzzled by the use of the test in Figs 3b and d, Figs 4b and d and Fig 8E, where I think more than 2 values may have been compared (or at least, it might be more appropriate to compare more than 2 values). Can the authors check and if necessary, redo the statistics with an ANOVA and suitable post-hoc test?
2) In 2021 the authors demonstrated that GB83 was an allosteric agonist at the PAR1 (Seo Y, Heo Y, Jo S, Park SH, Lee C, Chang J, Jeon DK, Kim TG, Han G, Namkung W. Novel positive allosteric modulator of protease-activated receptor 1 promotes skin wound healing in hairless mice. Br J Pharmacol. 2021 Sep;178(17):3414-3427. doi: 10.1111/bph.15489. Epub 2021 May 14. PMID: 33837955.) I cannot see a citation to this work. Whilst it is a different receptor, I think it is useful background information that readers would find useful.
3) The authors offer no suggestions as to the mechanism of GB83. Do they have any thoughts as to how it might be working? It is interesting that it always seem to produce a slower response than the other agonists.
Author Response
We greatly appreciate the editor’s and reviewers’ efforts to carefully review our manuscript and the valuable comments and suggestions offered for the improvement of the manuscript (ijms-1902488). We have made each of the suggested revisions. The points of criticism raised by the reviewers were addressed by a point-by-point response. Changes in the manuscript text are highlighted in red color font.
Reviewer 1
This paper shows, that contrary to previous reports in the literature, GB83 is a rapidly inactivating agonist at the PAR2. The data is novel and convincing, although the statistical analysis may need attention. I feel the paper will be of interest, given that the compound is routinely described as an antagonist in the literature.
Comments:
- The authors state that they use a paired t-test. This is appropriate for most of their experiments, where they are simply comparing two variables; the t-test cannot be used to compare three or more values. However, I am puzzled by the use of the test in Figs 3b and d, Figs 4b and d and Fig 8E, where I think more than 2 values may have been compared (or at least, it might be more appropriate to compare more than 2 values). Can the authors check and if necessary, redo the statistics with an ANOVA and suitable post-hoc test?
Response: Thank you for the helpful comment. The authors reconstructed statistics by ANOVA followed by Tukey’s post-hoc test for Figure 3B, 3D, 4B, 4D and 8E.
- In 2021 the authors demonstrated that GB83 was an allosteric agonist at the PAR1 (Seo Y, Heo Y, Jo S, Park SH, Lee C, Chang J, Jeon DK, Kim TG, Han G, Namkung W. Novel positive allosteric modulator of protease-activated receptor 1 promotes skin wound healing in hairless mice. Br J Pharmacol. 2021 Sep;178(17):3414-3427. doi: 10.1111/bph.15489. Epub 2021 May 14. PMID: 33837955.) I cannot see a citation to this work. Whilst it is a different receptor, I think it is useful background information that readers would find useful.
Response: Thank you for the comment. The previous study is briefly described in the introduction section.
- The authors offer no suggestions as to the mechanism of GB83. Do they have any thoughts as to how it might be working? It is interesting that it always seems to produce a slower response than the other agonists.
Response: As the reviewer comments, the authors also think that GB83 produces a slower response than trypsin and PAR2-AP. As shown in Figure S1, when intracellular calcium signaling via PAR2 activation by PAR2-AP and GB83 was observed at the single cell level, the increase in intracellular calcium level induced by GB83 was similar to that of PAR2-AP. However, PAR2-AP induced PAR2 activation simultaneously in most cells, whereas GB83 induced asynchronous PAR2 activation. Although this study did not elucidate why GB83 induces asynchronous PAR2 activation, this phenomenon may be attributed to the pharmacological properties of GB83, such as membrane permeability and solubility, or the binding mechanism between GB83 and PAR2 receptor. This is described in the revised discussion section.
Reviewer 2 Report
In their study the authors compare the properties of GB83 and PAR2-AP, a small molecule and peptide agonist of PAR2, respectively and conclude that these compound elicit distinct activation and regulation pathways.
Figure 1. In HT-29 cells expressing endogenous receptors, the calcium response to GB83 is markedly delayed compared to that elicited by trypsin and PAR2-AP (partly explaining the apparent antagonistic effect of GB83 on PAR2-AP). In Figure 2, however, the authors show that the GB83 stimulation of calcium is much faster, closer to that that elicited by trypsin and PAR2-AP in HT-29 cells. The authors should comment these findings. What is the potential contributing role of a much higher level of expressed PAR2 in stable A2058-PAR2 cells?
Figure 3-4. Consistent with slower calcium signaling, phospho-ERK and phospho-p38 signaling is also les efficient and delayed. Is it only because GB83 is partial agonist? How to explain the sustained phospo-p38 response in GB83-stimulated cells?
Figure 5-6. Trypsin and GB83 share, compared to PAR-2AP, the capacity of inducing sustained PAR2 endocytosis and PAR2 co-localization with beta-arrestins. However, in Figure 8 the recovery of PAR2 activation after washing is decreased and delayed in GB83 cells compared to trypsin and PAR2-AP treated cells. The authors should comment these findings.
In Figure 8, the recovery from trypsin-induced endocytosis in necessarily dependent of PAR2 repopulation from internal stores or after neo-synthesis (since the enzyme-activated receptor is degraded). On the other hand, recovery after PAR2-AP stimulation might be due to receptor recycling after beta-arrestin release. What is the mechanism of PAR2 recovery after GB83 stimulation? The authors should investigate the relative mechanisms involver in PAR2 recovery for the 3 agonists (using for example inhibitors of protein synthesis, brefeldin A to block repopulation from internal stores…)
Specific points
It would be useful to know the receptor density in cells expressing endogenous versus exogenous PAR2
Line 262. GB83 does not induce “stronger” of “faster” internalization of PAR2. Its effect in this regard is similar to that of trypsin. Just the functional recovery is slower (due to unknown mechanisms).
Line 271. Co-localization studies cannot determine the strength or the stability of the interaction with beta-arrestins
Line 275 It is not clear what are the mentioned beta-arrestin-mediated pathways.
Line 283. So far there is no evidence that the recovery of PAR2 from trypsin and GB83 stimulation depend on the same mechanism. In any case, the formation of a stable complex with beta-arrestin is clearly not the major mechanism that explains the delay of PAR2 recovery after trypsin-induced PAR2 degradation.
Author Response
We greatly appreciate the editor’s and reviewers’ efforts to carefully review our manuscript and the valuable comments and suggestions offered for the improvement of the manuscript (ijms-1902488). We have made each of the suggested revisions. The points of criticism raised by the reviewers were addressed by a point-by-point response. Changes in the manuscript text are highlighted in red color font.
Reviewer 2
In their study the authors compare the properties of GB83 and PAR2-AP, a small molecule and peptide agonist of PAR2, respectively and conclude that these compounds elicit distinct activation and regulation pathways.
Major comments:
- Figure 1. In HT-29 cells expressing endogenous receptors, the calcium response to GB83 is markedly delayed compared to that elicited by trypsin and PAR2-AP (partly explaining the apparent antagonistic effect of GB83 on PAR2-AP). In Figure 2, however, the authors show that the GB83 stimulation of calcium is much faster, closer to that that elicited by trypsin and PAR2-AP in HT-29 cells. The authors should comment these findings. What is the potential contributing role of a much higher level of expressed PAR2 in stable A2058-PAR2 cells?
Response: Thank you for the helpful comment. Since the expression level of PAR2 is higher in A2058-PAR2 cells than in HT-29 cells, GB83 can bind PAR2 receptors in PAR2-overexpressing cells much faster. When we further observed the effect of PAR2-AP on the increase of intracellular calcium in A2058-PAR2 cells, as shown in Figure 2B, PAR2-AP increased the intracellular calcium level faster than GB83 in A2058-PAR2 cells. Therefore, this phenomenon is likely due to increased expression of the PAR2 receptor. This is described in the revised discussion section.
- Figure 5-6. Trypsin and GB83 share, compared to PAR2-AP, the capacity of inducing sustained PAR2 endocytosis and PAR2 co-localization with beta-arrestins. However, in Figure 8 the recovery of PAR2 activation after washing is decreased and delayed in GB83 cells compared to trypsin and PAR2-AP treated cells. The authors should comment these findings.
Response: Thank you for the helpful comment. Comments on the findings were described in the revised discussion section.
- In Figure 8, the recovery from trypsin-induced endocytosis in necessarily dependent of PAR2 repopulation from internal stores or after neo-synthesis (since the enzyme-activated receptor is degraded). On the other hand, recovery after PAR2-AP stimulation might be due to receptor recycling after beta-arrestin release. What is the mechanism of PAR2 recovery after GB83 stimulation? The authors should investigate the relative mechanisms involver in PAR2 recovery for the 3 agonists (using for example inhibitors of protein synthesis, brefeldin A to block repopulation from internal stores)
Response: Thank you for the helpful comment. We observed the effect of brefeldin A on the recovery of PAR2 from GB83, PAR2-AP and trypsin-induced endocytosis in HT29 cells expressing EGFP-tagged PAR2. As shown in Figure S2, brefeldin A strongly blocked all the recovery from GB83, PAR2-AP and trypsin-induced endocytosis. These results suggest that the recovery of PAR2 from GB83-induced endocytosis is mainly due to the neo-synthesis of PAR2 rather than receptor recycling after beta-arrestin release.
Specific points
- It would be useful to know the receptor density in cells expressing endogenous versus exogenous PAR2
Response: We tried to determine the PAR2 receptor density by immunoblotting with three commercially available antibodies, but unfortunately, no reliable PAR2 antibody was found so far, so the experiment was not successful.
- Line 262. GB83 does not induce “stronger” of “faster” internalization of PAR2. Its effect in this regard is similar to that of trypsin. Just the functional recovery is slower (due to unknown mechanisms).
Response: Thank you. The sentence has been corrected.
- Line 271. Co-localization studies cannot determine the strength or the stability of the interaction with beta-arrestins.
Response: The authors agree with the reviewer's comments. The sentence has been corrected.
- Line 275 It is not clear what are the mentioned beta-arrestin-mediated pathways.
Response: Thank you. The sentence has been corrected.
- Line 283. So far there is no evidence that the recovery of PAR2 from trypsin and GB83 stimulation depend on the same mechanism. In any case, the formation of a stable complex with beta-arrestin is clearly not the major mechanism that explains the delay of PAR2 recovery after trypsin-induced PAR2 degradation.
Response The authors agree with the reviewer's comments. The sentence has been corrected.
Round 2
Reviewer 2 Report
Unfortunately the brefeldin A experiment is not contributive, since surface PAR2 recovery after trypsin, PAR2-AP or GB83 was similarly inhibited. We will never know why the functional recovery after GB83 is delayed and blunted compared to trypsin and PAR2-AP. Similarly there is no explanation of the so-called asynchronous PAR2 activation by GB83. I think that the absence of mechanistic elucidation of these findings reduces the overall interest of the study, but it will be the choice of the authors and of the editor to let these issues unsolved.
Minor points
The sentence” Downstream of these signaling pathways leading to the production of inflammatory cytokines initiates the PAR2-mediated inflammatory axis [8,9] line 40” seems to miss a subject, please reword/correct.
Please define what is PAR1-AP the first time it is mentioned in the text. I’m not sure that every potential reader knows what this peptide is.